# Investigation of Hot Deformation Behavior and Microstructure Evolution of Lightweight Fe-35Mn-10Al-1C Steel

Alexander Yu. Churyumov [1],*[ID], Alena A. Kazakova [1], Andrey V. Pozdniakov [1], Tatiana A. Churyumova [2] and Alexey S. Prosviryakov [1][ID]

1   Department of Physical Metallurgy of Non-Ferrous Metals, National University of Science and Technology MISiS, Leninskiy Prospekt 4, 119049 Moscow, Russia; m1805037@edu.misis.ru (A.A.K.); pozdniakov@misis.ru (A.V.P.); pro.alex@mail.ru (A.S.P.)
2   Joint-Stock Company "Advanced Research Institute of Inorganic Materials Named after Academician A.A. Bochvar", Rogova Str. 5a, 123098 Moscow, Russia; tachuryumova@bochvar.ru
*   Correspondence: churyumov@misis.ru; Tel.: +7-495-955-0134

**Abstract:** The deformation behavior of lightweight Fe-35Mn-10Al-1C steel with an elevated concentration of Mn was investigated. Hot compression tests at temperatures of 950–1150 °C and strain rates of 0.1–10 s$^{-1}$ were carried out using the thermomechanical simulator, Gleeble 3800. Strain compensated constitutive model of hot deformation behavior with high accuracy (error was 4.6%) has shown significant increases in the effective activation energy (410–460 kJ/mol) in comparison with low Mn steels. The significant influence of the strain rate and temperature on the grain size was shown. The grain size decreases from the initial value of $42 \pm 6$ μm to the value of $3.5 \pm 0.7$ μm after the deformation at 1050 °C and 10 s$^{-1}$. The model of the microstructure evolution of the investigated steel was constructed. The average error of the constructed model was 8.5%. The high accuracy of the constructed models allows for their application for the optimization of the hot deformation technologies using finite element simulation.

**Keywords:** hot deformation; microstructure modeling; thermomechanical simulator Gleeble; light-weight steel; constitutive model

## 1. Introduction

The development of new materials and technologies is an essential requirement to decrease the non-positive influence of human vehicles on the environment. Lightweight steel with a high strength may give such an opportunity. The Fe-Mn-Al-C alloying system is one of the prospective Fe-based materials with enhanced specific strength. These alloys may have a strength of more than 2 GPa [1] and may be applied as materials for automobiles, trains, and other transport vehicles. In most of the steels, the content of Mn is limited by 30 wt.%. At the same time, the increase in the alloying elements' content may increase the strength of the steel. However, such materials may lose technological plasticity, which may decrease the efficiency of their production [2].

The manufacturing of massive parts made from the Fe-Mn-Al-C usually includes such a process as hot deformation. The main cast defects such as liquation, porosity, and coarse grain microstructure may be significantly revealed during this process and preliminary heating [3–5]. However, the large material and energy losses may proceed in the case of the non-optimal parameters of thermomechanical treatment [6]. The determination of the hot deformation behavior, constitutive modeling of the flow stress, and microstructure evolution may help to optimize the hot deformation technologies. Currently, a large number of the constitutive models that connect the stress and thermomechanical parameters such as temperature, strain rate, and strain have been developed [7–10]. The values of the constants in the models are determined using the experimental data. Pernis has developed a simple methodology for the calculation of the constants in one of the types of constitutive models,

such as the Garofalo equation [11]. The constitutive models may be used for the finite element simulation of the industrial processes to provide the minimization of the energy costs and obtain the fine-grain microstructure [12–16].

Up to now, there are few models that have been constructed for the hot deformation behavior and grain microstructure evolution of lightweight Fe-Mn-Al-C steels. Zhang et al. developed strain-dependent equations for the Fe-23Mn-2Al-0.2C twinning-induced plasticity steel [17]; Yang et al. have determined the dependence between the Zener–Hollomon parameter and flow stress during the hot deformation of Fe-27Mn-11.5Al-0.95C steel [18] and constructed the model for the dynamic recrystallization [19]. Renault et al. have constructed the model for deformation and fracture behavior for Fe-30Mn-10Al-1.1C steel with Mo addition [20]. The processes of dynamic recrystallization (DRX) and precipitation in the Fe-11Mn-10Al-0.9C lightweight steel were investigated by Liu et al. [21]. Wu et al. have shown an increase in the Al content from 8 to 10 wt.% in Fe-26Mn-xAl-1C steel significantly increases the activation energy of the hot plastic deformation [22]. However, most of the investigations of lightweight hot deformation behavior were devoted to steels with an Mn content lower than 30 wt.%.

The purpose of this study was to investigate and model the flow stress and microstructure evolution during the hot deformation of the lightweight steel with an elevated Mn content.

## 2. Materials and Methods

The investigated steel namely, Fe-35Mn-10Al-1C, has the following composition: Fe-35.1Mn-9.8Al-1.05C-0.2Si-0.05Cr-0.04Ni-0.01P-0.005S-0.005N-0.002O (wt.%). The Fe-35Mn-10Al-1C steel cylinders with a diameter of 6 mm and a length of 60 mm were obtained using raw materials of commercial purity by melting in a vacuum induction furnace Indutherm 20V in argon atmosphere followed by casting to the copper mold. The cylinders were cut to samples with a height of 9 mm for the compression using a Gleeble 3800 thermomechanical simulator (DSI Inc., New York, NY, USA). The scheme of the test is shown in Figure 1. The specimens were heated at a rate of 5 °C/s to 1150 °C, held for 30 s, and compressed to a true strain of 0.1 with a strain rate of 1 s$^{-1}$ followed by the annealing for 60 s. After the preliminary thermomechanical treatment, the samples were cooled for the main deformation stage. The compression in the temperature range of 950–1150 °C was made to a true strain of 1 at a constant true strain rate of 0.1, 1, and 10 s$^{-1}$. The tantalum and graphite foils were used to decrease the influence of the friction between the sample's edges and the anvils. The true stress-strain curves were recalculated to consider the adiabatic heating and friction during the deformation accordingly [23,24].

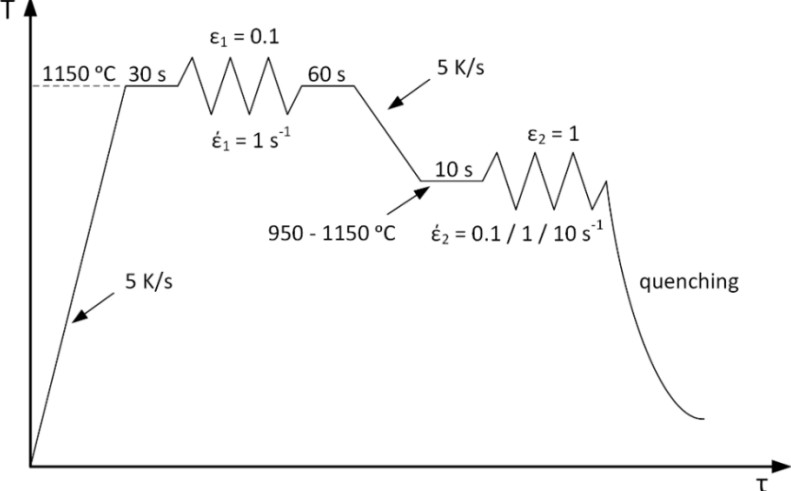

**Figure 1.** Scheme of the compression test.

A Tescan-VEGA3LMH scanning electron microscope (SEM) was used for the microstructural investigations and energy-dispersive X-ray spectroscopy analysis (EDS) (TESCAN, Brno, Czech Republic). The samples were mechanically polished and chemically etched in a 5% nital solution. The grain size was determined by the linear intercept method by treating three pictures for each state. The phase diagram and Scheil curve for the Fe-35Mn-10Al-1C steel were obtained by the thermodynamic calculations using the Thermocalc program with the TCFe7 database.

## 3. Results

### 3.1. Initial Microstructure and Phase Composition of the Fe-35Mn-10Al-1C Steel

The as-cast microstructure of the investigated steel consists of elongated grains with a dendritic structure (Figure 2a). The EDS mapping has shown that most of the elements except the carbon are distributed homogenously (Figure 2c). The presence of the carbide phase is a result of non-equilibrium crystallization. As one can see in the temperature dependence of the solid phases mass fraction calculated using the Scheil model, firstly, bcc-(Fe) crystalizes from the liquid state (Figure 3a). The peritectic reaction L+bcc-(Fe)→fcc-(Fe) is almost suppressed, and the crystallization of the austenite begins at 1268 °C. The last crystallization reaction is L→fcc-(Fe)+$M_5C_2$. The two-stage deformation scheme (Figure 1) was applied to eliminate the non-equilibrium dendritic structure and make the main compression in the one-phase austenite range. As one can see in Figure 2b, the microstructure of the steel consists of equiaxed austenite grains with an average size of 42 ± 6 μm. According to the calculated phase diagram (Figure 3a), the hot compression has proceeded in the austenite phase region.

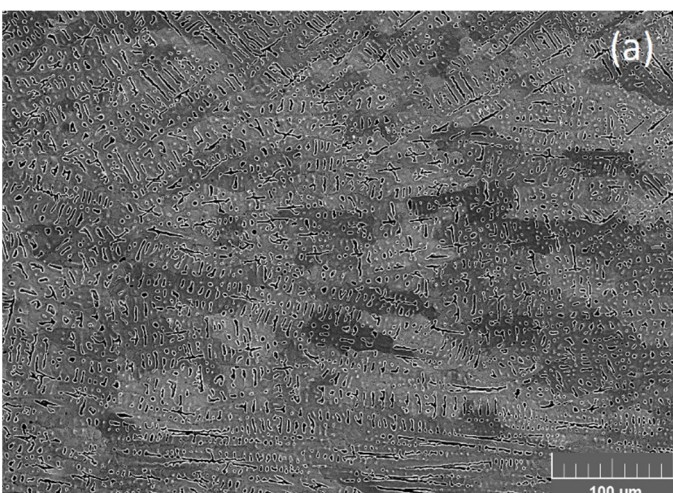
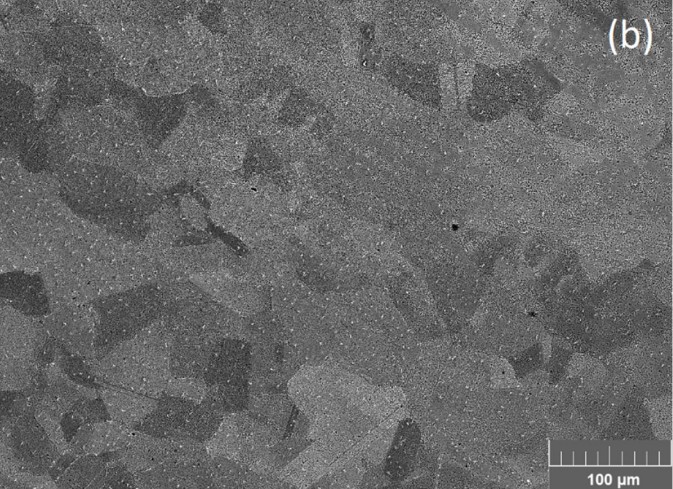

**Figure 2.** *Cont.*

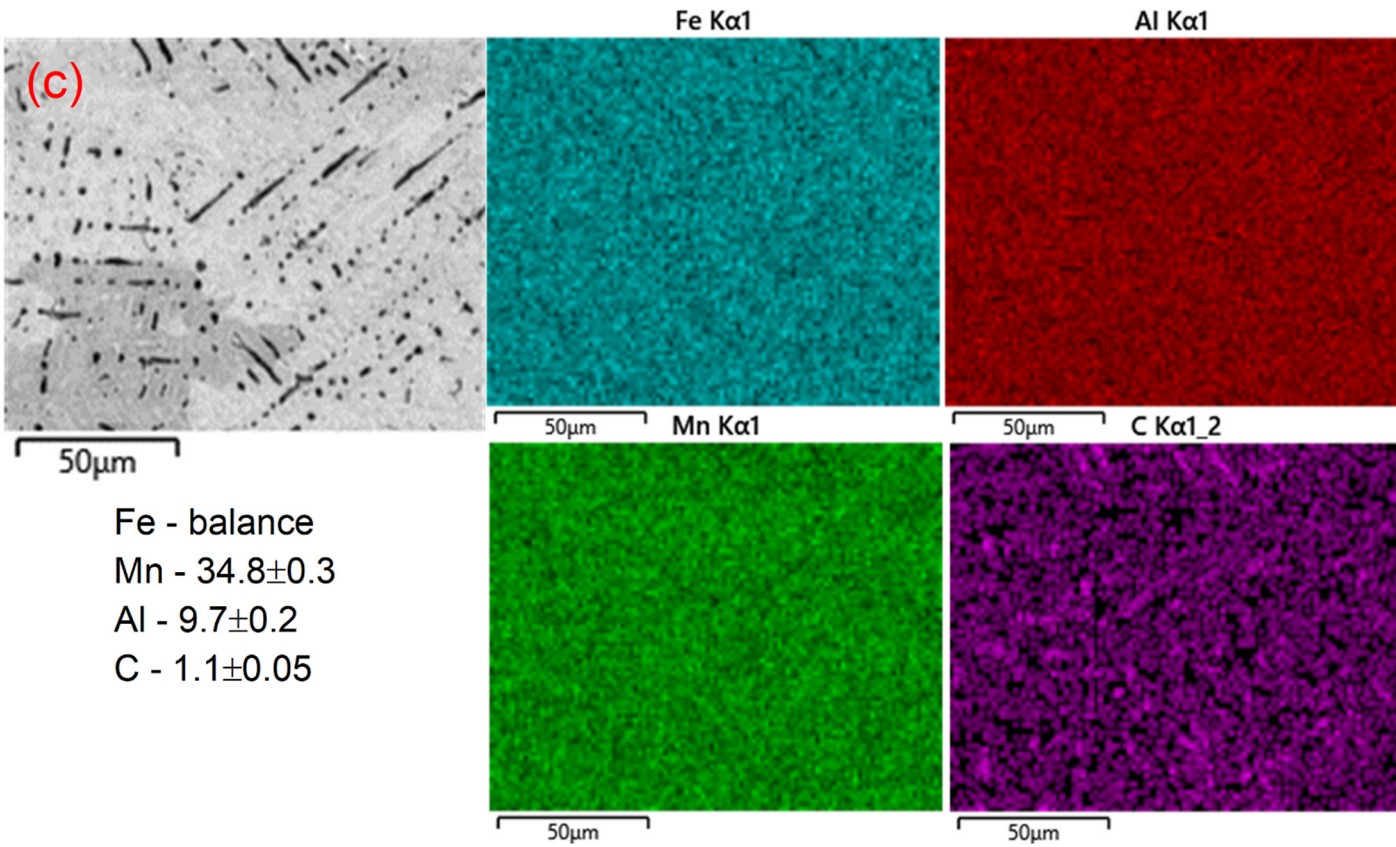

**Figure 2.** Microstructure of the investigated steel: in the as-cast state (**a**) and before the main deformation stage (**b**). The mapping and the chemical composition of the selected zone in the as-cast sample (**c**).

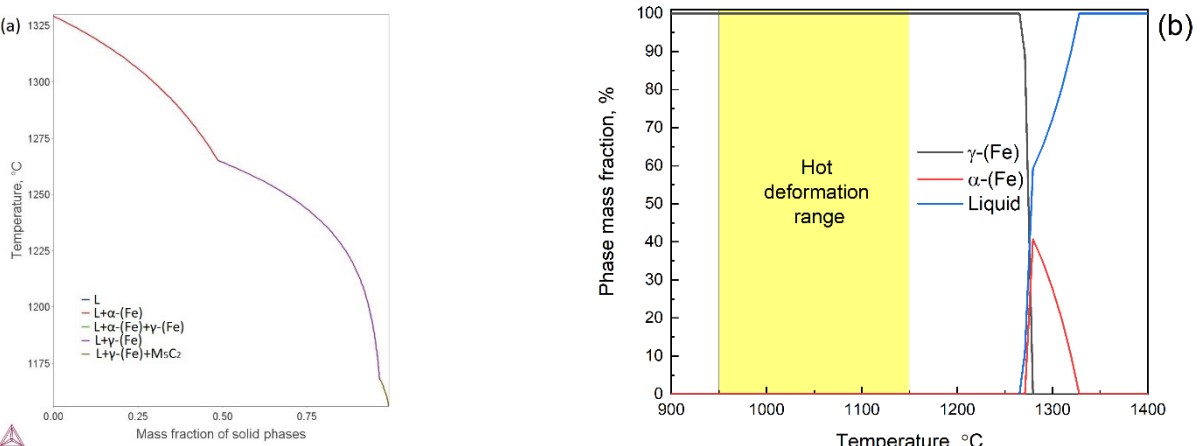

**Figure 3.** The calculated mass fraction of solid phases (Scheil model) (**a**) and high-temperature phase composition (**b**) of the Fe-35Mn-10Al-1C steel.

### 3.2. Hot Deformation Behaviour of the Fe-35Mn-10Al-1C Steel and Constitutive Modeling

As seen in Figure 4, Fe-35Mn-10Al-1C steel shows typical hot deformation behavior. At the first moment of the compression, the stress increases with attenuation due to concurrence between strain hardening and dynamic recovery. After the achievement of the critical strain, the dynamic recrystallization (DRX) begins to contribute to the softening that provides the maximum seen on the stress-strain curves. After the smooth decline, the stress becomes constant, which corresponds to the steady stage of the deformation. The stress

has a lower value for the higher temperatures and lower strain rates for all compression conditions as a result of the diffusion's decisive role in hot deformation processes.

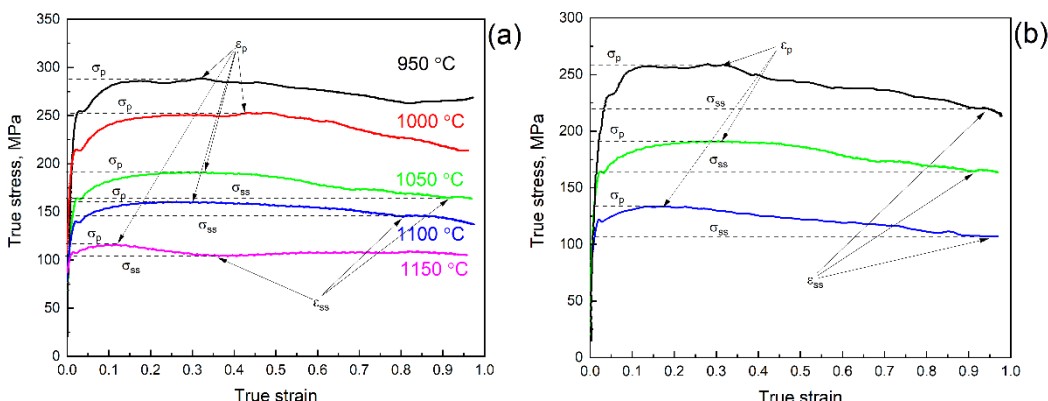

**Figure 4.** Hot compression curves for the Fe-35Mn-10Al-1C steel at a strain rate of 1 s$^{-1}$ (**a**) and a temperature of 1050 °C (**b**). The peak stress ($\sigma_p$), peak strain ($\varepsilon_p$), steady-state stress ($\sigma_{ss}$), and steady-state strain ($\varepsilon_{ss}$) are indicated by arrows.

The constitutive strain-dependent model was constructed to quantitatively describe the hot deformation behavior of Fe-35Mn-10Al-1C steel. The model is based on the relation between the true stress ($\sigma$) and Zener–Hollomon parameter ($Z$) [25]:

$$Z = \dot{\varepsilon}e^{\frac{Q}{RT}} \tag{1}$$

where $\dot{\varepsilon}$, $T$, $Q$, and $R$ are strain rate (s$^{-1}$), temperature (K), effective activation energy (J/mol), and the universal gas constant (8.314 J/mol·K), respectively. The hyperbolic sine dependence is universal for the description of the true stress dependence on the deformation conditions:

$$Z = A_3[\sinh(\alpha\sigma)]^{n_2} \tag{2}$$

where $A_3$, $n_2$, and $\alpha$ are constants of the material. The description of partial cases of the constitutive models is needed to determine the $\alpha$ parameter's value. The power law may be applied for low stresses:

$$Z = A_1\sigma^{n_1} \tag{3}$$

The exponential form well describes the conditions when a high level of stress may be achieved:

$$Z = A_2e^{\beta\sigma} \tag{4}$$

where $A_1$, $n_1$, $A_2$, and $\beta$ are material parameters that may be determined using the experimental values of the stress.

Parameter $\alpha$ may be calculated as:

$$\alpha \approx \frac{\beta}{n_P} \tag{5}$$

All the material's constants were determined by the linearization of Equations (2)–(4) and following minimization between the calculated and experimental stress values using the least square method. It is known that the material's constants are mostly dependent on the strain [26,27]. As a result, the values of the constants were determined for the strain values of 0.05; 0.1; 0.2 … 0.9. The dots in Figure 5 show the values of the constants at different strains. The effective activation energy $Q_3$ for the investigated steel has a value in the range of 410–460 kJ/mol. In addition, the constitutive model was constructed for the peak stresses. A comparison between calculated and experimental values is shown in Figure 6e. The effective activation energy has a value of 432 kJ/mol. The $Q_3$ values are

larger than for the similar steels with a lower Mn content. So, the effective activation energy has a value of 294.2 and 355.5 kJ/mol for the Fe-27Mn-11.5Al-0.95C [19] and Fe-11Mn-10Al-0.9C steels [21], respectively. Two factors may influence the deformation: the higher concentration of the solid solution and the presence of the two phases in the microstructure. In the case of the investigated steel, the higher Mn concentration significantly decreases the movement of the dislocations and inhibits the softening processes such as dynamic recovery and recrystallization.

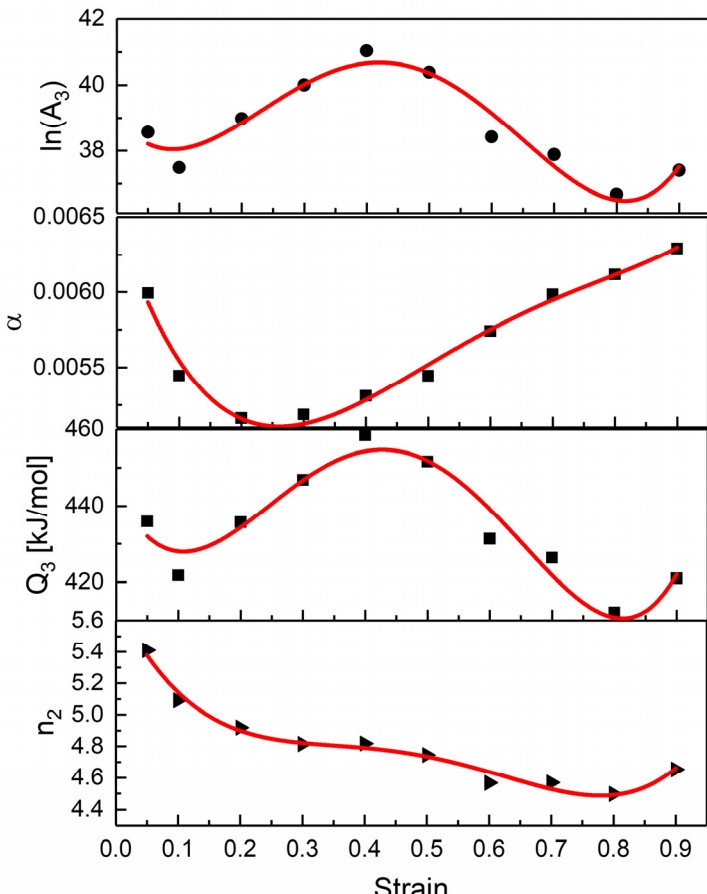

**Figure 5.** The dependence of the constitutive model coefficients on the true stress: dots are calculation values; lines are the polynomial approximation accordingly Equations (6)–(9).

The constants $A_3$, $\alpha$, and $Q_3$ are insignificantly changed in the range of about $\pm 5\%$ of the average values. However, coefficient $n_2$ decreases more noticeably with increases in the strain. Usually, the value of this constant is reciprocal to the strain rate sensitivity coefficient. The increase in this coefficient shows that the materials flow stabilizes at the higher values of the strain due to refinement of the grain s by the DRX.

The fourth-order polynomials were used to mathematically describe the dependences of the material's constants on the strain (lines in Figure 5):

$$ln(A_3) = 39 - 24.1\varepsilon + 177.9\varepsilon^2 - 347\varepsilon^3 + 196.7\varepsilon^4 \qquad (6)$$

$$\alpha = 0.0065 - 0.013\varepsilon + 0.039\varepsilon^2 - 0.044\varepsilon^3 + 0.017\varepsilon^4 \qquad (7)$$

$$Q_3 = 444 - 338.3\varepsilon + 2171.5\varepsilon^2 - 4055\varepsilon^3 - 2255.5\varepsilon^4 \qquad (8)$$

$$n_2 = 5.72 - 8.19\varepsilon + 27.74\varepsilon^2 - 41.18\varepsilon^3 + 21.14\varepsilon^4 \qquad (9)$$

A comparison between the calculated stress values and experimental curves is presented in Figure 6a–c. As seen, the model shows good accordance and may be used for the

stress calculation for other deformation conditions which may be useful in the case of the finite element simulation using this material.

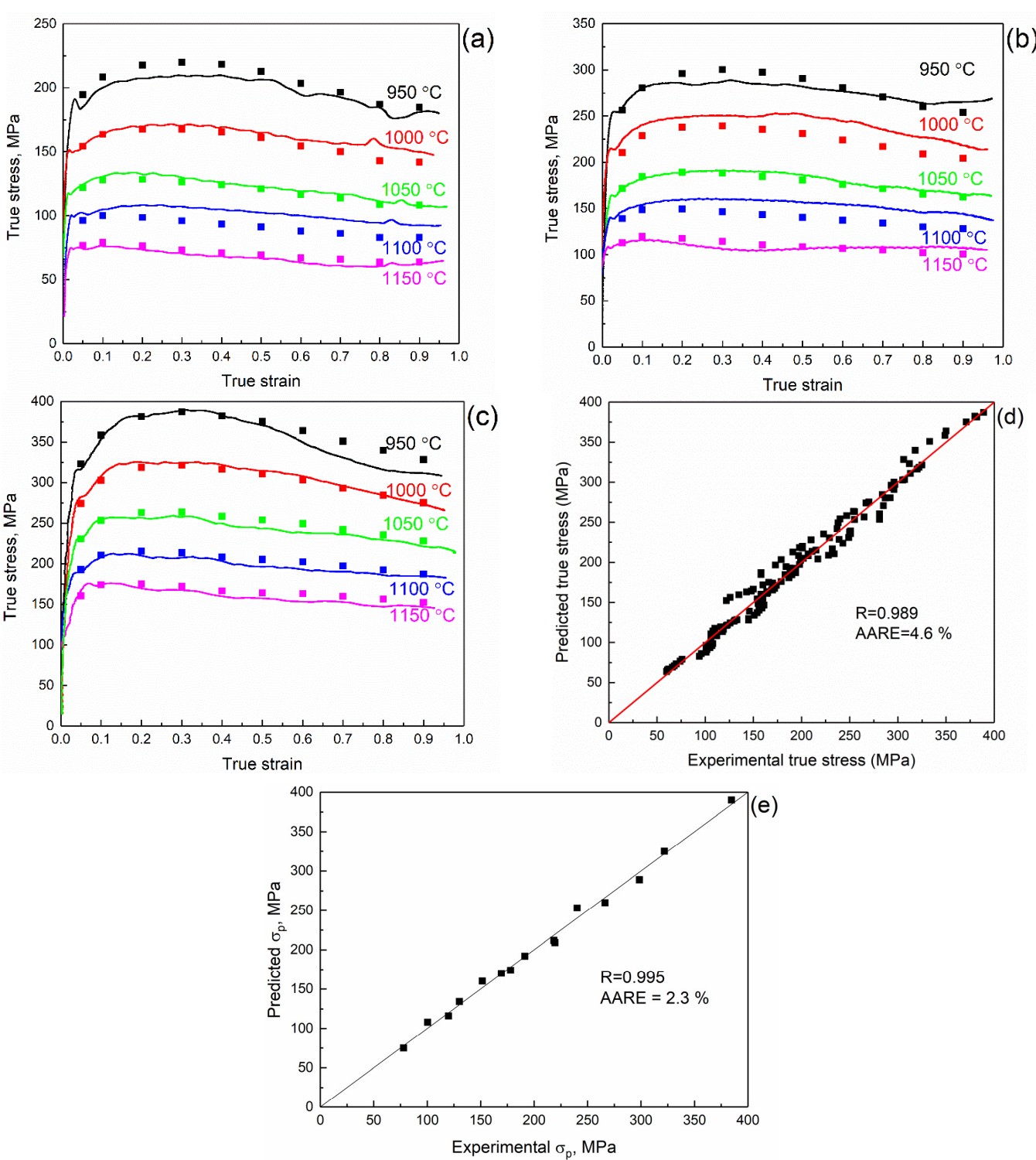

**Figure 6.** A comparison between the experimental and predicted true stress for the Fe-35Mn-10Al-1C steel at a strain rate of 0.1 s$^{-1}$ (**a**), 1 s$^{-1}$ (**b**), 10 s$^{-1}$ (**c**), for the whole dataset (**d**) and for peak stress values (**e**).

The accuracy of the obtained constitutive model was estimated using Pearson′s correlation coefficient (*R*) and average absolute relative error (*AARE*), which are expressed as [28]:

$$R = \frac{\sum_{i=1}^{N}\left(E_i - \overline{\overline{E}}\right)\left(P_i - \overline{P}\right)}{\sqrt{\sum_{i=1}^{N}\left(E_i - \overline{\overline{E}}\right)^2 \sum_{i=1}^{N}\left(P_i - \overline{P}\right)^2}} \tag{10}$$

$$AARE(\%) = \frac{100}{N}\sum_{i=1}^{N}\frac{|E_i - P_i|}{E_i} \tag{11}$$

where *E* and *P* are the experimental and predicted stresses. $\overline{\overline{E}}$ and $\overline{P}$ are average values. *N* is the number of experimental values. As shown in Figure 6d, the model′s accuracy is good (*R* = 0.989 and *AARE* = 4.6%). The adequacy of the constructed model was also checked using the additional experiments and Fisher′s criterion. The calculations in Appendix A have shown the adequacy of the model.

The deformation conditions may be significantly changed during real industrial processes. The constructed model may help to predict the magnitude of the stress in such a situation. It is known that the temperature may significantly decrease due to natural cooling. As a result, the value of the true stress increases (Figure 7a). Such a situation may lead to the equipment or tools breakdown. The decrease in the strain rate may help to prevent the breaking due to its influence on the stress level (Figure 7b).

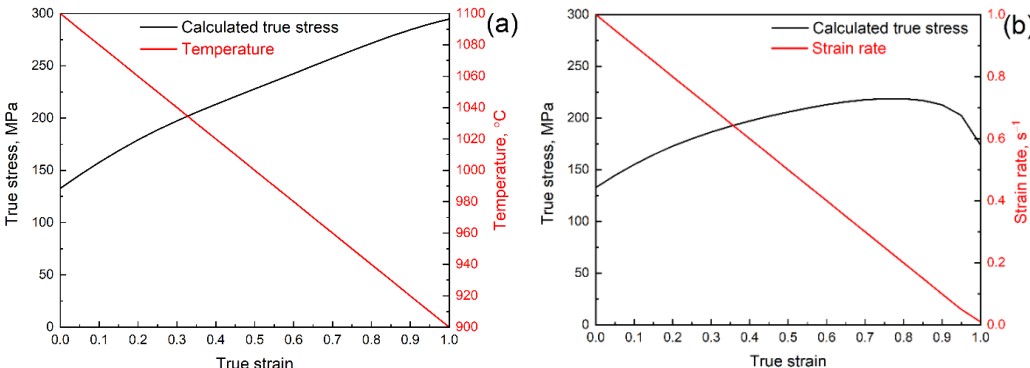

**Figure 7.** The calculated true stress values in the case of the changing of the temperature at a constant strain rate of 1 s$^{-1}$ (**a**) and in the case of the changing of the strain rate at the same temperature profile (**b**).

### 3.3. Microstructure Evolution during the Hot Deformation

The hot deformation significantly influenced the grain microstructure of the investigated steel (Figure 8). The deformation at temperatures higher than 1050 °C provides a fully recrystallized microstructure in contrast with the microstructure after the deformation at a lower temperature that consists of a large initial unrecrystallized grains and small DRX grains (Figure 9). The DRX grain size increases with increases in the temperature and decreases in the strain rate (Figure 10a). The grain after the deformation at 1050 °C and 10 s$^{-1}$ refines more than ten times in comparison with the initial value. At lower temperatures, the DRX process did not finish, and we can see in the microstructure elongated coarse initial grains and groups of fine crystallites.

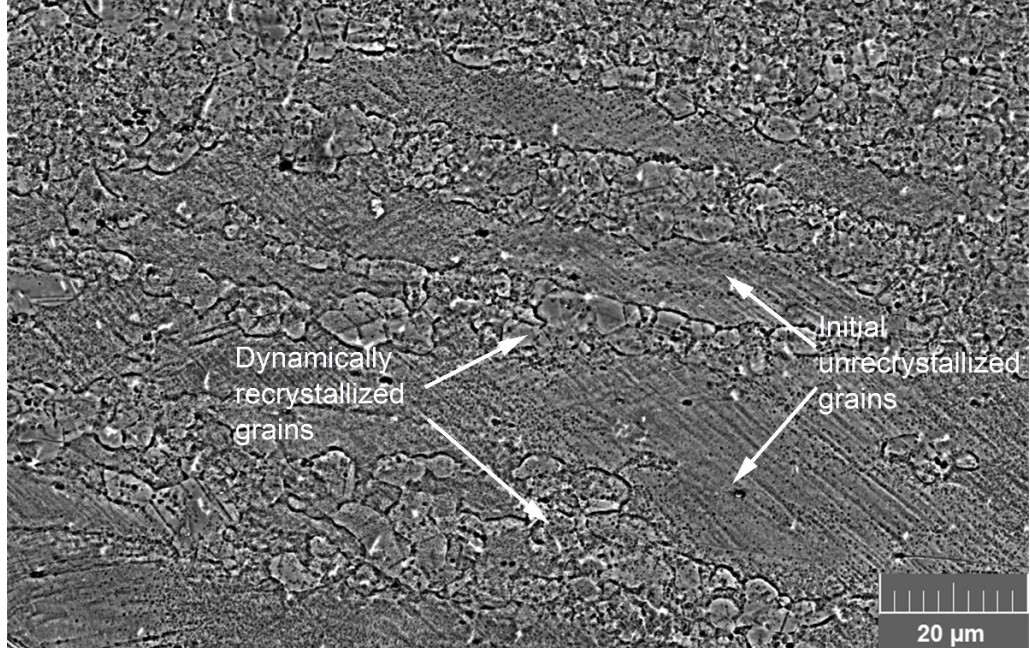

**Figure 8.** The microstructure of the investigated steel after the hot deformation.

**Figure 9.** The partially recrystallized microstructure after the deformation at 950 °C and 10 s$^{-1}$.

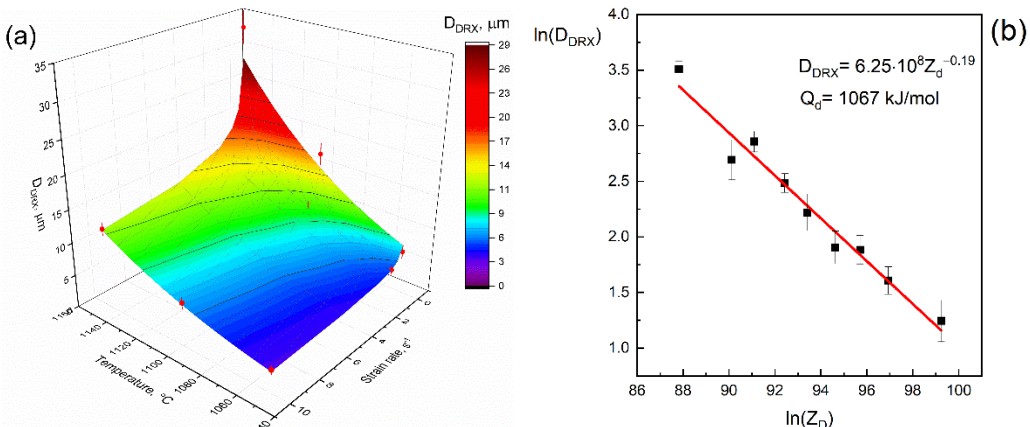

**Figure 10.** The dependence of dynamically recrystallized grain size on the deformation temperature and strain rate (**a**). The ln($D_{DRX}$) vs. $Z_D$ plot (**b**).

The values of the dynamically recrystallized grains obtained at the temperatures of 1050–1150 °C were used to construct the microstructure evolution during the DRX. The size of the DRX grains at the steady-state stage is determined by the strain rate and deformation temperature in terms of the Zener–Hollomon parameter:

$$D_{DRX} = a_d \cdot Z_d^m \tag{12}$$

$$Z_d = \dot{\varepsilon}\exp\left(\frac{Q_d}{R \cdot T}\right) \tag{13}$$

where $a_d$, $m$, $Q_d$ are material constants whose values were found by the multilinear regression analysis of the logarithm form of Equation (12) using the experimental values of the grain size obtained at the temperatures of 1050–1150 °C and the strain rate of 0.1–10 s$^{-1}$. The dependence $ln(D_{DRX})$ vs. $ln(Z_d)$ is shown in Figure 10b. A comparison between the experimental and predicted values of the grain size is shown in Figure 10a. As seen, the power low between grain size and the Zener–Hollomon parameter shows good accuracy.

The microstructure of the investigated steel shows a higher strain rate sensitivity in comparison with other steels. The coefficient $m$ has a value of −0.19. The usual value of the coefficient for the steels is in the range of −0.167−−0.136 [29–31]. The large value of the coefficient provides the possibility of significant refinement of the grain microstructure by increasing the strain rate that raises the mechanical properties.

**4. Conclusions**

1. The hot deformation behavior and microstructure evolution of the lightweight Fe-35Mn-10Al-1C steel with an elevated concentration of the Mn were investigated and modeled.
2. It was shown that the as-cast austenite/ferrite microstructure of the steel transforms to the one-phase austenite equiaxed grains after the deformation on a true strain of 0.1 and annealing for 60 s.
3. The high-accuracy constitutive model has shown that the effective activation energy of the investigated steel (410–460 kJ/mol) has a larger value than for the similar lightweight steels with a lower Mn content.
4. The grain microstructure of the steel is significantly influenced by hot deformation conditions. The grain size decreases from the initial value of 42 ± 6 μm to the value of 3.5 ± 0.7 μm after the deformation at 1050 °C and 10 s$^{-1}$.
5. Constructed constitutive models may be applied in finite element simulation to develop the hot deformation technologies for new Fe-35Mn-10Al-1C steel with an elevated concentration of Mn.

**Author Contributions:** Conceptualization, A.Y.C.; methodology, A.Y.C.; validation, A.Y.C. and T.A.C.; investigation, A.A.K., A.S.P., A.V.P.; resources, A.Y.C.; data curation, T.A.C.; writing—original draft preparation, A.Y.C., A.V.P.; supervision, A.Y.C.; project administration, A.Y.C.; funding acquisition, A.Y.C. All authors have read and agreed to the published version of the manuscript.

**Funding:** This research was funded by the Russian Science Foundation (project №18-79-10153-P).

**Institutional Review Board Statement:** Not applicable.

**Informed Consent Statement:** Not applicable.

**Data Availability Statement:** Data are contained within the article.

**Conflicts of Interest:** The authors declare no conflict of interest.

## Appendix A

The additional experiments were carried out to check the adequacy of the constructed model using Fisher's criterion. First of all, the reproducibility dispersion ($S_y^2$) was determined using the following formula:

$$S_y^2 = \sum_{i=1}^n \frac{S_i^2}{n},\tag{A1}$$

where dispersion $S_i^2$ was determined for each strain (0.05; 0.1; 0.2 ... .0.9) using the three stress-strain curves obtained at the same deformation temperature and strain rate (Figure A1a):

$$S_i^2 = \frac{\sum_{j=1}^m \left(y_{ij} - \overline{y_i}\right)^2}{m}\tag{A2}$$

where $y_{ij}$ and $\overline{y_i}$ are the value of the stress at each strain for the different curves and its average values. m is a number of the curves. $S_y^2 = 78.2$.

Secondly, the adequacy dispersion ($S_{ad}^2$) was determined using additional experiments at lower temperatures and higher strain rates (for extrapolation ability of the obtained model) and intermediate temperature and strain rate (for the interpolation) (Figure A1b):

$$S_{ad}^2 = \frac{\sum_{j=1}^k \left(y_j^{calc} - y_j^{pred}\right)^2}{k}\tag{A3}$$

where $y_j^{calc}$ and $y_j^{pred}$ are calculated and predicted values of the stress. $k$ is a number of points for checking ($k = 30$). $S_{ad}^2 = 132.5$.

The calculated value of Fisher's criterion was determined using the following equation:

$$F^{calc} = \frac{S_{ad}^2}{S_y^2}\tag{A4}$$

Obtained value $F^{calc} = 1.69$ was compared with a table value $F^{tab} = 1.86$. $F^{calc} < F^{tab}$, as a result, the constructed model is adequate.

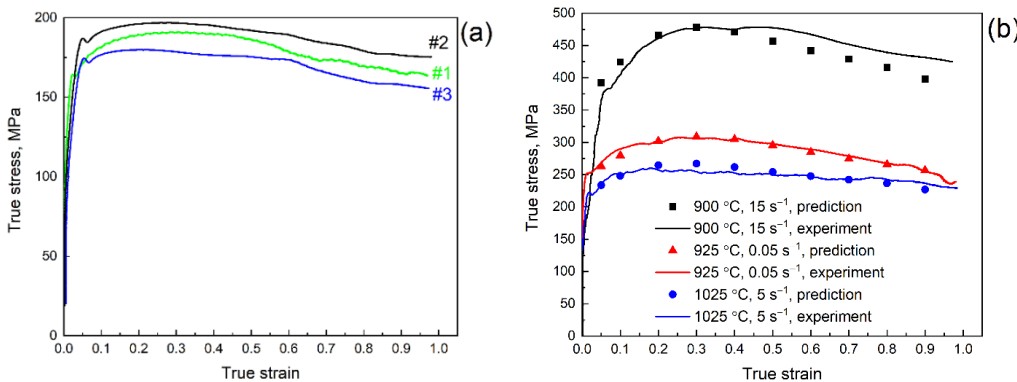

**Figure A1.** The additional stress-strain curves obtained for determination of the (**a**) reproducibility dispersion ($S_y^2$) and (**b**) adequacy dispersion ($S_{ad}^2$).

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
