# Peer review of "Investigation of Hot Deformation Behavior and Microstructure Evolution of Lightweight Fe-35Mn-10Al-1C Steel"

_metals, doi:10.3390/met12050831_

Round 1
Reviewer 1 Report
In this paper, the hot deformation behavior and microstructure evolution of the lightweight Fe‐35Mn‐10Al‐1C steel with an elevated concentration of the Mn were investigated. Some revison should be performed:
(1) Some images in Fig. 7 are not clear to show the grains. The obvious scratches can be seen. More careful sampling should be made to better show the grains.
(2) The authors stated that "The main cast defects such as liquation, porosity, and coarse grain microstructure may be significantly revealed during this process, and preliminary heating". What does “reveal” mean?The expression is not so clear. Please explain it.
Author Response
Dear Reviewer!
Thank you for your thorough consideration of our paper “Investigation of Hot Deformation Behavior and Microstructure Evolution of Lightweight Fe-35Mn-10Al-1C steel”, your kind response, and valuable comments. The authors have tried to answer your questions.
Please, find attached file with the replies.
Reviewer 2 Report
Paper is trying to address hot deformation constitutive models for high Mn steel. Choice of material is interesting; however, some issues need addressing prior paper being suitable for publication.
- Choice of thermo-mechanical program is peculiar. Although, later in the text explanation is given for use for prior deformation at soaking temperature, whole setup is very unusual. At 1150 °C and true strain 0.1 you are already in recrystallization according to given flow curves. Moreover, why not normalize samples prior to hot compression tests to eliminate dendritic microstructure?
- Experimental setup of hot compression test is not given in enough detail. How was dealt with friction between sample and anvil.
- Constitutive procedure for determination of apparent activation energy for deformation is valid. However, usually activation energy is determined either for peak stress or steady state stress. Reason for this is in time needed for processes at hand where at higher strain rates generally higher strain is needed to cause recrystallization. Please explain why activation energy was calculated for the same true strains for each of three strain rates used.
- Please explain what method was used to determine grain size. Also micrographs are too dark to distinguish features. For initial microstructure, elemental distribution would be appreciated to reveal any Mn segregations.
Author Response
Dear Reviewer!
Thank you for your thorough consideration of our paper “Investigation of Hot Deformation Behavior and Microstructure Evolution of Lightweight Fe-35Mn-10Al-1C steel”, your kind response, and valuable comments. The manuscript was modified accordingly to your advice.
Please, find attached file with the replies.

Reviewer 3 Report
- Chap.1: In analysis of models describing dependence between ZH parameter and flow stress and strain rate is missing Garofalo equation e.g. given at: [R. Pernis: Simple methodology for calculating the constants of Garofalo equation, Acta Metallurgica Slovaca, Vol. 23, 2017, No. 4, p. 319-329, DOI10.12776/ams.v23i4.1017p ]
- Chap.2: Analysis of precise local chemical composition is missing.
- Chap.2:...steel ingots with a diameter of 6 mm and a length of 60 mm... The term "ingot" means a large geometric dimension, so the authors have to change the term "ingot".
- Chap.2: The holding time of 30 s at a reheating temperature of 1150 ° C for a sample diameter of 6 mm is not sufficient for to obtained a homogeneous thermal field.
- Fig.3b: The existence of the α- (Fe) phase at a temperature more than 1320 ° C is impossible!
- Fig.4: The start of DRX recrystallization is not clear from the graphical dependencies, even the curve obtained at 1 s-1 and T = 1150 ° C had a multi-peak character. The authors have to mark on each curve the points with axes (φ_peak,σ_peak) and (φ_SS,σ_SS).
- Fig.5: It is not clear from the text on the based of which the graphic dependencies were obtained! Also, numerical descriptions of dependencies by polynomials of fourth order are not convenient, because the data calculated at inter - node points are incorrect.
- Fig.7: The micrographs are very poor quality and without confirmation of the DRX recrystallized microstructures. The etching solution was not selected correctly.
- Fig.8: Based on the previous statements, the validity of this figure is not clear.
Author Response

(The authors gave the same response as above.)

Reviewer 4 Report
This is the article where the authors proposed a model for calculating stress-strain curves (SSC) as well as microstructure parameters during hot deformation of Fe-35Mn-10Al-1C steel. The article is well written, and its results are of practical interest, since the developed model can be used in FEM to calculate the instantaneous stress values during plastic deformation. The reviewer would recommend the article for publication after the addressing following concerns:
1. Authors should indicate whether they took into account deformation heating when calculating stresses. And if not, then it is necessary to estimate the error.
2. In total, 15 SSC are published in the article. To test the adequacy of the model, it is not enough to show only the general indicator of the quality of the approximation of experimental data (AARE=4.6%). The authors should use part of the curves to determine the coefficients of the model, and the other part to check the adequacy of the model. Moreover, both interpolation and extrapolation would be of interest.
3. In addition, it is of interest to analyze the sensitivity of the Fe-35Mn-10Al-1C steel to changes in parameters. As is known, real processes are not carried out at a constant strain rate and a constant temperature. If these parameters change, then it can significantly affect the magnitude of the stresses. A theoretical analysis or an experiment of the effect should be added.
4. Do the authors consider that the grain size is also the initial parameter for stress calculation. In the results in Fig. 8, they show that the grain size decreases with deformation. In the case of a real multi-pass rolling or forging process calculated by FEM, it is necessary to take into account both grain size reduction due to deformation and grain size grow due to static and dynamic recrystallization processes in order to know what grain size will be in the next pass. But the reviewer does not see et all the effect of grain size on stress in Eqs (2)-(9).
Author Response

(The authors gave the same response as above.)

Round 2
Reviewer 2 Report
Thank you for reply. Most issues were correctly answered, but the reasoning why activation energy was calculated for the same true strains for each of three strain rates used is hollow.
Reasoning to make polynomial dependence does not make scientific sense. If there would at least be some comparison to either steady state or peak apparent activation energy that might be acceptable. So please expand discussion on this section.
Moreover, justification for the use of prior deformation at soaking temperature of 1150 °C and true strain 0.1 could be accepted. However, if you soak at this temperature for example 180 s, would you expect different grain size in different samples? It would be beneficial to know obtained grain size prior to hot deformation.
In the industrial ingot hot deformation, normalization is usually performed prior to any deformation. After ingot stripping, you need to heat it up to required temperature where normalization is usually combined in this step.
Author Response
Thank you for the consideration of our revised paper “Investigation of Hot Deformation Behavior and Microstructure Evolution of Lightweight Fe-35Mn-10Al-1C steel”, your kind response, and valuable comments. The manuscript was modified accordingly to your advice. Please, find the replies in attached file.

Reviewer 3 Report
- Chap.2: Analysis of precise local chemical composition is always missing. Authors have to do local spectral analysis of chemical composition with definition minimally 10 elements given in mass % , including carbon content, oxygen and nitrogen.
- Fig. 10: The authors have to explain from where the value of the activation energy DRX Qd=1067 kJ/mol.
Author Response

(The authors gave the same response as above.)

Reviewer 4 Report
I have read the new version of the article, and I think that the authors have significantly improved it and I recommend it for publication.
Author Response
The authors thanks the Reviewer for the kind opinion about our paper.